# Usage of Digital Health Tools and Perception of mHealth Intervention for Physical Activity and Sleep in Black Women

**DOI:** 10.3390/ijerph19031557

**Published:** 2022-01-29

**Authors:** Yue Liao, Kyrah K. Brown

**Affiliations:** Department of Kinesiology, College of Nursing and Health Innovation, University of Texas at Arlington, Arlington, TX 76019, USA; kyrah.brown@uta.edu

**Keywords:** wearables, behavioral intervention, health promotion, behavioral change strategies

## Abstract

Black women of reproductive age are disproportionately affected by chronic health conditions and related disease risk factors such as physical inactivity and obesity. Health promotion studies need to focus on this population since optimizing preconception health will lead to improvement of both birth outcomes and the woman’s long-term health. mHealth interventions that utilize wearable sensors to provide personalized and timely feedback can be used to promote physical activity (PA). This study aimed to examine Black women’s wearable sensor usage and their perceptions about future mHealth interventions that target PA and sleep. Our analysis included 497 Black women (aged 18–47 years) who completed a cross-sectional online survey. Ninety-two percent of participants did not meet the recommended level of PA, and 32.3% reported poor sleep quality. More participants indicated interest in a remote PA program (77%) than an in-person one (73%). More than half of participants indicated interest in receiving personalized feedback messages based on wearable trackers about PA (58.1%) and sleep (63.5%). This perceived acceptability of remote intervention and wearable-based feedback messages did not differ by socioeconomic status. Remotely delivered mHealth interventions that utilize wearables can be a viable behavioral change strategy to promote PA and sleep quality in Black women.

## 1. Introduction

In the United States, the prevalence of cardiovascular disease (CVD) risk factors among women of reproductive age (18–49) has demonstrated a 40% increase over the last two decades, with the prevalence substantially greater among non-Hispanic Black women (hereafter referred to as Black women) [1]. Consequently, an increasing number of Black women are entering pregnancy with common CVD risk factors such as obesity, diabetes, and hypertension [1,2]. Further, compared to other racial groups, a greater percentage of Black women of reproductive age have undiagnosed CVD risk factors [2] which often go unrecognized and untreated [3,4]. Nonpregnant Black women of reproductive age are an important population for CVD prevention research considering that over half of all pregnancies are unplanned and the implications of cardiovascular health on maternal health (which is also an important marker of the overall health of a nation) and health later in life [5]. For example, CVD is not only the leading cause of death among Black women [6]; CVD, including cardiomyopathy, accounts for 27.8% of preventable maternal deaths among reproductive-age Black women [7]. However, compared to pregnant and older women (≥50 years), nonpregnant women of reproductive age are largely underrepresented in cardiovascular health research [4,8,9].

Several factors can increase one’s risk of developing CVD, including non-modifiable factors, such as genetics and age, and modifiable behavioral factors, such as smoking, diets rich in saturated fat and cholesterol, physical inactivity, and inadequate sleep [10]. While we recognize all these factors have their unique contributions to a person’s CVD risk, this paper will focus on two of the important modifiable behavioral factors, physical activity and sleep, in the scope of implications for digital and mobile health (mHealth) interventions. Physical activity and sleep represent two daily behaviors that can be automatically and continuously tracked by a wearable sensor, thus, providing opportunities to develop personalized, adaptive behavioral interventions [11].

Available evidence indicates that the disparity in some of these behavioral factors, such as physical activity and sleep, in Black women of reproductive age contributes to the disparities in cardiovascular health [12,13]. According to national surveillance data, Black women of reproductive age, compared to other racial/ethnic groups, are less likely to report getting the recommended levels of physical activity (defined as getting at least 150 min per week of moderate-intensity activity or at least 75 min per week of vigorous-intensity activity) and less likely to report being normal weight (defined as having a body mass index of 18.5–24.9 kg/m^2^) [14]. Further, compared to white women of reproductive age, Black women report shorter sleep prevalence regardless of pregnancy status [15]. A higher prevalence of Black reproductive age women also reports having trouble falling asleep, trouble staying asleep, and waking up most days (i.e., ≥4 days in the previous week) not feeling rested [15].

There is growing recognition that the fundamental driver of persistent racial/ethnic disparities in CVD, as well as disparities in behaviors such as physical activity and sleep, is structural racism [16]. Structural racism refers to the totality of ways in which society fosters racial discrimination through mutually reinforcing systems of housing, education, employment, healthcare, and other social determinants [16,17]. As a recognized determinant of health, structural racism creates stressful and unhealthy social and built environments which, in turn, can contribute to disparate behavioral risk factors, such as physical inactivity and inadequate sleep, among Black women [16,18]. For example, several social and structural factors are associated with lower levels of physical activity among Black women, including residential racial segregation, resulting in communities with fewer physical characteristics that promote physical activity [19,20], perceived time availability and monetary cost of exercise facilities, family/caregiving responsibilities, and lack of a physical activity partner or role models [21]. Similarly, decades of sleep disparities research indicate that a number of neighborhood and environmental factors, such as residential racial segregation (which can transcend socioeconomic status), and neighborhood ambient features of urban environments, such as loud noises or bright lights, can be deleterious to sleep [22]. Racial discrimination also has a known effect on sleep health through the consequences of psychosocial trauma, discrimination, and microaggressions [23]. Self-reported experiences of racial discrimination have been associated with shorter sleep duration among Black women [24]. It is believed that those experiencing racism—interpersonally or at the structural level—may have lower quality sleep because of greater “racism-related” vigilance—an inability to set aside worry and stress due to lifelong experiences with discrimination [22]. 

In attempts to address disparities in cardiovascular health, behavioral interventions that aim to promote an active lifestyle and adequate sleep are considered as some of the effective strategies to reduce CVD risk in Black women. Digital and mHealth interventions, which leverage mobile phones, wearable sensors, and related, Internet-linked services for health communication and monitoring, present innovative opportunities to promote more equitable health outcomes among Black women of reproductive age [25,26]. Current evidence suggests that consumer-based, wearable sensors and smartphones may serve as viable and feasible interventions to promote physical activity and sleep among reproductive-age Black women [27,28]. In the U.S., a greater percentage of Black women own smartphones (80%) compared to White women (72%) [29]. In addition, a slightly greater percentage of Black Americans (23%) reported using smartwatches or fitness trackers compared to all U.S. adults (21%) and White Americans (20%) [30]. Prior research has demonstrated high acceptability among Black Americans regarding the use of smartphones, smartwatches, or fitness tracker data for health research [26,30]. Despite high levels of access and acceptability of consumer-based, wearable sensors and smartphones among Black women, they remain underrepresented in mHealth research [26,31]. For example, Joseph and colleagues [31] identified only three digital and mHealth physical activity interventions, published between 2000 and 2016, that focused exclusively on Black women (i.e., [21,32,33]). One systematic review identified a few additional intervention studies that included a predominately Black sample (i.e., [34,35,36]). Joseph and colleagues also reported that those studies primarily used Internet-based websites and text messages; none reported evaluating the use of wearable sensors to promote physical activity. There are also few published mHealth studies focused on the use of consumer-based, wearable sensors to promote sleep among Black women [28]. Black women of reproductive age are well positioned to benefit from mHealth interventions using consumer-based, wearable sensors [26]. For example, utilizing data from consumer-facing, wearable sensors is an innovative way to provide personalized feedback to increase daily physical activity levels and decrease time in sedentary activities [11,37]. Recent advancements in wearable sensors also make tracking sleep much more accessible than before [38,39], thus, giving us a complete picture of a person’s 24 h movement behaviors (i.e., physical activity, sedentary behavior, and sleep). More research, however, is needed to inform the development of tailored, wearable, sensor-based mHealth intervention strategies that promote physical activity and sleep among Black women. 

The purpose of this study is to explore Black women’s current wearable sensors usage and perceptions of receiving personalized feedback from those wearable sensors. The objectives of this study are to: (1) describe the current physical activity level and sleep quality in Black women of reproductive age; (2) assess their current usage and perceptions of wearable sensors; and (3) explore the associations between participants’ demographic characteristics and wearable sensors usage/perceptions. The present study contributes to the growing literature demonstrating the immense need for wearable sensor mHealth interventions focused on physical activity and sleep quality in racial/ethnic minority populations, specifically Black women [28,40].

## 2. Materials and Methods

The current study used a cross-sectional online survey of Black/African American women of reproductive age (18–49) who are residing in the United States. Participants were recruited using Prolific Academic, an online research platform designed to connect academic researchers with people who are interested in participating in online research studies [41]. Prolific Academic has been widely used as a high-quality platform for crowdsourcing behavioral research and provides a diverse pool of participants [42]. Once the research team posted the study survey to Prolific Academic, prospective participants who met the eligibility criteria were identified and notified by Prolific. Eligible individuals clicked the study survey link, completed the informed consent, and, if they agreed to the study, completed the survey. Participants were compensated at the rate of USD 9.54 per hour to complete this survey study.

QuestionPro (QuestionPro, Seattle, WA, USA) an online survey software, was used to administrate this survey study. The survey included 46 questions that assessed participants’ current physical activity level, sleep quality, health information technology usage, wearable tracker usage, perceptions of remotely delivered intervention, and demographic information. Survey items were derived from existing, validated survey instruments or from previous national studies [43,44,45]. Since the primary purpose of the survey was not to obtain a complete picture of participants’ behaviors (e.g., usual physical activity and sleep patterns), we used items from validated surveys to capture a “snapshot” of their current physical activity and sleep quality while keeping the survey brief. Please see the Appendix A for our survey.

The survey study was posted to Prolific Academic in February 2021 and the total response was capped to 500 individuals. Eligible participants enrolled in the study on a first-come, first-served basis. Data collection was completed within one week. Participants, on average, took 7 min to complete the survey. This study was approved by the Institutional Review Board at the University of Texas at Arlington (protocol #: 2021–0207). 

Statistical analyses were conducted using the SPSS version 26.0 (IBM Corp, Armonk, NY, USA). Descriptive statistics were generated for all variables, including the mean and standard deviation for continuous variables and percentages for categorical variables. Binary logistic regression was used to obtain both the crude and adjusted estimated differences in responses by demographic characteristics. The adjusted models included all demographic variables in one model. A *p*-value of 0.05 or less was considered significant.

## 3. Results

### 3.1. Participant Characteristics

A total of 499 females enrolled and completed the survey study. Two individuals were excluded from the analysis since they did not indicate themselves to be Black or African American in our demographic question. Thus, our analytical sample included 497 Black women. Their average age was 28 years old (SD = 7.50, ranged 18–47 years). The majority of the women were non-Hispanic (95.9%), US-born (95.8%), and not currently pregnant (97.4%). Thirty-three percent of the women had been pregnant and/or had delivered a child. Table 1 details their demographic information. 

### 3.2. Physical Activity Level and Sleep Quality

More than half of the women (54.3%) reported that they engaged in less than 30 min of moderate-intensity activities per week, 26.4% reported between 30–90 min/week, 11.5% reported between 90–150 min/week, and 7.8% reported more than 150 min/week. Eighty-two percent of the women indicated that they “need to be more active”. Thirty-seven percent of the participants indicated that they planned to start exercising regularly in the next 30 days, and 18% indicated they would start in the next 6 months.

Women in this study reported having an average of 6.6 h of sleep (SD = 1.38). Thirty-two percent of the women reported to have “fairly bad” or “very bad” sleep quality, and 27.2% indicated they had taken medication (prescribed or over-the-counter) to help them sleep in the past month. 

### 3.3. Digital Health Tools Usage and Perceptions of mHealth Interventions

Thirty-nine percent of the women reported to have owned a wearable fitness tracker or smartwatch, and 16% reported that they have not owned one but were planning to get one in the next few months. Table 2 shows the results from the crude and adjusted logistic regression model. Compared to those who were not employed, participants who were employed full-time were more likely to have owned a wearable device in both the crude (odds ratio = 2.241; *p* < 0.05) and adjusted models (odds ratio = 1.835; *p* < 0.05). Those who had completed a bachelor’s degree or higher were more likely to have owned a wearable device than those without a bachelor’s degree (51.2% vs. 30.8%, respectively; odds ratio = 2.359, *p* < 0.01). This effect of education level persisted after controlling for all other demographic variables (odds ratio = 1.897, *p* < 0.01). Those with an annual household income of more than $50,000 were more likely to have owned a wearable device than those with less income (49.3% vs. 31.1%, respectively; odds ratio = 2.157, *p* < 0.01). The effect of household income was consistent in the adjusted model (odds ratio = 1.852, *p* < 0.01). Wearable device ownership did not differ by age, marital status, insurance status, or chronic disease conditions. Consistent with results for current/previous ownership, participants who were employed full-time were more likely to report planning to get a wearable device in the next few months compared to those who were not employed in both the crude (odds ratio = 2.449; *p* < 0.05) and adjusted models (odds ratio = 2.390; *p* < 0.05). Further, women aged 30 years and older were more likely to report planning to get a wearable device compared to those who were younger (34.8% vs. 21.1%, respectively; odds ratio = 1.989, *p* = 0.01). This effect of age persisted after controlling for all other demographic variables (odds ratio = 2.016, *p* < 0.05). Planning to get a wearable device did not differ by marital status, education, household income, insurance status, or chronic disease conditions. The most frequently reported primary goal for using the wearable trackers was to track activity levels for both current/previous (35.8%) and planned owners (15.5%; see Figure 1). 

All but three women reported owning a smartphone (99.4%). Of those who owned a smartphone, 56.5% owned an iPhone, and 43.5% owned an Android phone. Eighty-eight percent of smartphone owners reported using their phones to look up information about a health condition during the past 12 months. Sixty-one percent reported they had used a health-related smartphone application (app) at least a couple times during the past 12 months. Fifteen percent reported using a health-related app almost daily (see Figure 2). Table 3 shows the results from the crude and adjusted logistic regression model. Women aged 30 years and older were less likely to have used any health-related app in the past 12 months compared to those who were younger (55.1% vs. 65.0%, respectively; odds ratio = 0.659, *p* = 0.03). The effect of age was consistent in the adjusted model (odds ratio = 0.543, *p* < 0.01). Compared to those who were uninsured, participants who had insurance were more likely to have used a health-related app in both the crude (odds ratio = 1.970; *p* < 0.05) and adjusted models (odds ratio = 2.003; *p* < 0.05). In the adjusted model, employment status (comparing full-time employed with non-employed) was associated with health-related app usage (odds ratio = 1.777, *p* < 0.05). Marital status (comparing single vs. other) was significant in the crude model only. Health-related app usage (yes vs. no) did not differ by education, household income, or chronic disease conditions. 

Nineteen percent of the women reported being interested in an in-person intervention program that helps them to become more active in their daily life. Another 54.5% reported being interested in such a program only when the COVID-19 pandemic is over. Seventy-seven percent of the women reported being interested in a remotely delivered program (such as through website, emails, text messages, smartphone apps) to help them become more active. Table 4 shows the results from the crude and adjusted logistic regression model. Marital status (comparing single vs. married) was significant in the crude model only. In the adjusted model, chronic disease conditions became a significant predictor of having an interest in remotely delivered physical activity program. Compared to those without any chronic disease conditions, those with at least one chronic disease condition were more likely to report having an interest in a remote physical activity program (odds ratio = 1.692, *p* = 0.04). Fifty-eight percent of the women reported being interested in receiving personalized feedback messages about their activity levels based on a wearable device. Fifty-eight percent of the women reported being interested in non-pharmaceutical approaches to improve their sleep. Sixty-four percent of the women reported being interested in receiving personalized feedback messages about their sleep based on a wearable device. The likelihood of willingness to receive wearable, device-based, personalized feedback messages about activity levels and sleep did not differ by any of the demographic characteristics.

## 4. Discussion

In this cross-sectional survey study, we recruited Black women of reproductive age from an online research platform (Prolific Academic) and assessed their current physical activity and sleep behaviors, as well as their usage and perceptions towards digital health tools and interventions. Black women in our study had a similar proportion of marital status and insurance coverage when compared to available national data for Black women of reproductive age [46,47]. Compared to national data, the proportion of our study participants with at least a four-year degree was higher (26.2% (national) vs. 42.5% (sample)) [48], and the proportion of the study participants with employment was lower (89.6% (national) vs. 62.8% (sample)) [49]. Importantly, this study captured responses from Black women of reproductive age who were mostly insufficiently active (92.2% not meeting the recommended physical activity level), reflecting a population that is in need of physical activity interventions. We believe the present study contributes to the dearth of cardiovascular health-related mHealth literature focused on Black women of reproductive age in the United States. There are four key findings from our study that have the potential to guide further research and development of mHealth interventions using consumer-based, wearable sensors to promote physical activity and sleep.

First, in addition to reporting being mostly insufficiently physically active, Black women in this study reported an average of 6.6 h of sleep, which is slightly below the recommended 7–9 h of sleep, on a regular basis [50]. Approximately one-third of the participants reported having bad sleep quality. These results are consistent with findings from national datasets of women of reproductive age and highlight racial/ethnic disparities in physical activity and sleep [14,15]. Although it was beyond the scope of the present study, our finding underscores the need for more research investigating the impact of racism on physical activity and sleep among Black women of reproductive age. Further, to ensure more equitable health outcomes in this population, more research is needed to ensure that mHealth interventions are not only culturally tailored, but also responsive to the social context of Black women, which is inextricably linked to structural racism [25,51,52].

Second, nearly all participants in the current study reported owning a smartphone. These data are consistent with the increasing trend in smartphone ownership nationwide [53] and studies that focused on Black women [26,29,30]. Further, the present study yielded results that can inform mHealth intervention development targeting Black women. For instance, 88% of our study participants reported using their phones to look up information about a health condition, and almost two-thirds reported using a health-related app at least a couple of times in the past year, suggesting a good amount of familiarity with smartphone-based, digital health tools. Among study participants who owned or planned to own a wearable fitness tracker or smartwatch, the primary goal for using the device was to track their activity levels. Moreover, 58% and 64% of study participants indicated that they would be interested in receiving personalized feedback based on a wearable tracker regarding their physical activity level and sleep, respectively. This finding is consistent with prior studies, with community-based samples reporting that between 64% and 69% of Black women ages 18–50 were willing to participate in research using wearable trackers [26,27]. Our findings provide support for the utilization of wearable sensors, in particular consumer-facing wearables, as an excellent intervention tool for Black women. 

Third, more than three-quarters (77%) of the study participants expressed an interest in remotely delivered physical activity interventions. This level of interest is higher than for an in-person program, even after being asked not to consider the potential concerns over the COVID-19 pandemic (73%). This high perceived acceptability of remote interventions is consistent with the recent findings that individuals are now very knowledgeable of the virtual environment, partly due to the COVID-19 pandemic that pushed people to work, learn, and socialize online [54]. Notably, many existing digital and mHealth interventions still involve some in-person components (e.g., for baseline assessment, brief educational sessions). It would be interesting for future studies to explore if interventions can be effectively conducted in a completely virtual/online environment (e.g., from participant onboarding and baseline assessment to follow-up assessment). 

Finally, the present study identified important variations in digital health tools usage and perceptions among reproductive-age Black women. There is a growing recognition that racial/ethnic group data disaggregation is necessary to ensure health equity in racial/ethnic minority populations. In other words, examining within-group differences among Black women of reproductive age can provide crucial insights that can ensure the development of effective interventions [55,56]. An earlier study examining Black women’s willingness to participate in mHealth research identified important differences in perceptions and behaviors by age [26]. The present study explored associations between digital health tools usage and perceptions with additional participant demographic characteristics. After adjustment, the likelihood of owning a wearable fitness tracker, sensor device, or smartwatch was greater among women who reported a bachelor’s degree or higher, being full-time employed, and having a household income of ≥$50,000. This finding suggests that researchers designing mHealth intervention studies with Black women should prepare to provide participants with wearable fitness trackers or smartwatches to eliminate the potential financial barriers. Additionally, the likelihood of using a health-related smartphone app was greater among women who were older (aged 30–49 years), full-time employed, and insured. It is possible that this finding is partly due to the increased adoption of electronic medical services by health providers (e.g., patient portals, such as MyChart) where individuals with insurance are more likely to have frequent encounters. Further, many employers now offer programs for employee wellness that take advantage of health-related apps and wearable sensors (e.g., team-based wellness challenges), thus, providing opportunities for employees to interact more with health-related apps. Lastly, after adjustment, the likelihood of expressing interest in remotely delivered interventions was higher among women with at least one chronic health condition. Although most study participants expressed an interest in remotely delivered interventions, this finding underlines the importance of potentially prioritizing women with existing chronic health conditions that may increase their risk for CVD in line with expectancy value theories in public health [57,58,59]. For example, it is plausible that women with chronic health conditions may have a higher level of motivation and expected value of such mHealth interventions. 

Despite the strengths of the present study, there are several potential limitations. First, the present study relied on self-report data which could have been subject to social desirability bias and recall bias. Future research should use both self-report measures of participant perceptions and objective measures for behavioral outcomes such as physical activity and sleep patterns. Second, all study participants were recruited using Prolific Academic, an online participant recruitment platform. Although Prolific Academic generally provides good quality data [42], the study sample represents reproductive-age Black women with Internet access who enrolled themselves into the study via Prolific. Thus, the results may not be generalizable to all Black reproductive age women in the United States. Nevertheless, we were able to recruit Black women from across the country with a wide range of education and income levels. Furthermore, since nearly all survey participants were insufficiently active, the opinions we gathered from them represent the same population that future physical activity interventions aim to target. 

## 5. Conclusions

Black women of reproductive age experience disparate physical activity and sleep patterns, which contribute to CVD risk. This is an urgent public health issue because of the negative effects of CVD on women’s pregnancy outcomes and health later in life. Despite their high levels of access to and use of smartphones and wearable sensors, Black women remain underrepresented in mHealth research related to cardiovascular health. This study contributed to an understanding of reproductive-age Black women’s current physical activity and sleep patterns, current digital health tool usage, and perceptions towards future mHealth interventions. Remotely delivered mHealth interventions that utilize personalized feedback could be a viable behavioral change strategy to promote physical activity and sleep quality in reproductive-age Black women. Researchers should ensure that intervention strategies are responsive to the needs of Black women subgroups that may be overlooked during intervention development (e.g., low-income and uninsured Black women, Black women with chronic health conditions, Black women of advanced maternal age (e.g., 35–49 years old), etc.). Doing so requires continued research studies that exclusively focus on Black women and include diverse sociodemographic variables. Future research studies could also obtain more in-depth information, such as motivation and acceptance of using digital health tools, types of information presented in the feedback messages (e.g., type of wearable sensor monitoring data, tones and presentations of health-related messages), through qualitive research methods such as focus groups. Future studies could further investigate the optimal content, timing, and frequency of feedback messages based on wearable devices, as well as opinions (e.g., perceived usefulness) towards existing digital health tools in this population.

## Figures and Tables

**Figure 1 ijerph-19-01557-f001:**
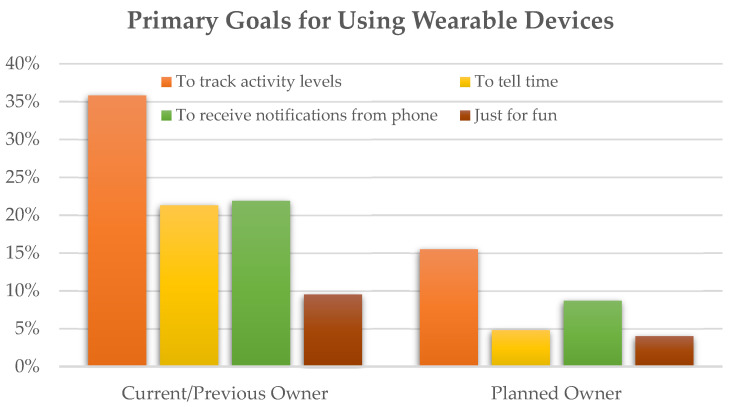
Primary goals for using wearable trackers by individuals who have owned a tracker and those who planned to get one in the next few months.

**Figure 2 ijerph-19-01557-f002:**
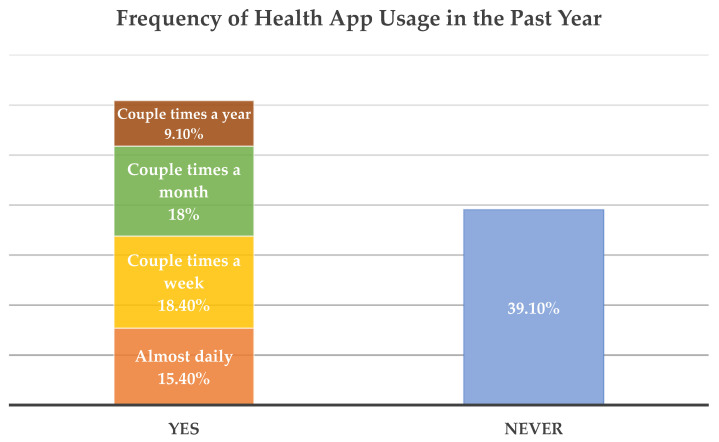
Frequency of using a health-related smartphone application in the past 12 months.

**Table 1 ijerph-19-01557-t001:** Participants’ demographic characteristics (n = 497).

Demographic Variable	N	Percentage
Marital Status		
Single	371	74.6%
Married	97	19.5%
Separated/Divorced/Widowed	25	5.0%
Prefer not to say/Missing	4	0.8%
Employment Status		
Full-time employed	173	34.8%
Part-time employed	92	18.5%
Self-employed	47	9.5%
Not employed	154	31.0%
Other	27	5.4%
Education Level		
High school diploma or lower	59	11.9%
Some college or vocational training	227	45.7%
Bachelor’s degree	149	30.0%
Master’s degree or higher	62	12.5%
Annual Household Income		
Less than $20,000	92	18.5%
$20,001–$35,000	93	18.7%
$35,001–$50,000	79	15.9%
$50,001–$75,000	85	17.1%
$75,001–$100,000	73	14.7%
Greater than $100,000	53	10.7%
Prefer not to say	22	4.4%
Insurance Status		
Insured	411	82.7%
Uninsured	64	12.9%
Not sure	22	4.4%
Chronic Disease Conditions		
None	308	62.0%
One	163	32.8%
More than one	26	5.2%

**Table 2 ijerph-19-01557-t002:** Ownership of wearables by demographic characteristics among Black women.

	Have Owned		Planned to Own	
	Odds Ratios (95% Confidence Interval)		Odds Ratios (95% Confidence Interval)	
	Crude	Adjusted *	Crude	Adjusted *
Age				
<30 years old (ref)	1.00	1.00	1.00	1.00
≥30 years old	1.169 (0.807–1.694)	0.915 (0.578–1.447)	**1.989** **(1.173–3.373)**	**2.016** **(1.068–3.805)**
Marital Status				
Single (ref)	1.00	1.00	1.00	1.00
Married	1.246 (0.792–1.960)	0.687 (0.387–1.222)	1.602 (0.852–3.011)	1.095 (0.501–2.396)
Other	1.223 (0.562–2.662)	1.130 (0.465–2.746)	0.700 (0.193–2.546)	0.601 (0.152–2.370)
Employment Status				
Not employed (ref)	1.00	1.00	1.00	1.00
Full-time employed	**2.241** **(1.430–3.512)**	**1.835** **(1.077–3.128)**	**2.449** **(1.236–4.849)**	**2.390** **(1.097–5.204)**
Other	0.940 (0.590–1.498)	0.989 (0.587–1.668)	1.878 (0.981–3.593)	1.626 (0.795–3.327)
Education				
Less than bachelor’s degree (ref)	1.00	1.00	1.00	1.00
Bachelor’s degree and higher	**2.359** **(1.632–3.411)**	**1.897** **(1.219–2.952)**	1.076 (0.628–1.844)	0.638 (0.318–1.281)
Household Income				
≤$50,000 (ref)	1.00	1.00	1.00	1.00
>$50,000	**2.157** **(1.482–3.140)**	**1.852** **(1.218–2.818)**	1.442 (0.845–2.459)	1.592 (0.850–2.984)
Insurance Status				
Uninsured (ref)	1.00	1.00	1.00	1.00
Insured	0.961 (0.562–1.643)	0.832 (0.452–1.529)	1.707 (0.718–4.058)	1.493 (0.559–3.984)
Chronic Disease Conditions				
None (ref)	1.00	1.00	1.00	1.00
At least one	1.131 (0.782–1.637)	1.177 (0.780–1.777)	1.066 (0.627–1.811)	0.942 (0.521–1.703)

* The adjusted models included all demographic variables listed in the table. Bold indicates *p* < 0.05.

**Table 3 ijerph-19-01557-t003:** Health-related smartphone application (app) usage * by demographic characteristics among Black women.

	Odds Ratios (95% Confidence Interval)	
	Crude	Adjusted ^+^
Age		
<30 years old (ref)	1.00	1.00
≥30 years old	**0.659** **(0.454–0.957)**	**0.543** **(0.343–0.860)**
Marital Status		
Single (ref)	1.00	1.00
Married	0.954 (0.601–1.516)	0.861 (0.486–1.523)
Other	**0.457** **(0.210–0.993)**	0.598 (0.253–1.416)
Employment Status		
Not employed (ref)	1.00	1.00
Full-time employed	1.548 (0.986–2.431)	**1.777** **(1.035–3.051)**
Other	1.096 (0.704–1.707)	1.124 (0.682–1.851)
Education		
Less than bachelor’s degree (ref)	1.00	1.00
Bachelor’s degree and higher	1.373 (0.950–1.986)	1.242 (0.792–1.945)
Household Income		
≤$50,000 (ref)	1.00	1.00
>$50,000	1.372 (0.944–1.995)	1.206 (0.789–1.843)
Insurance Status		
Uninsured (ref)	1.00	1.00
Insured	**1.970** **(1.159–3.349)**	**2.003** **(1.106–3.627)**
Chronic Disease Conditions		
None (ref)	1.00	1.00
At least one	1.133 (0.780–1.646)	1.265 (0.836–1.914)

* Have used at least a few times vs. never used in the past 12 months. ^+^ The adjusted models included all demographic variables listed in the table. Bold indicates *p* < 0.05.

**Table 4 ijerph-19-01557-t004:** Interest in remotely delivered physical activity programs * by demographic characteristics among Black women.

	Odds Ratios (95% Confidence Interval)	
	Crude	Adjusted ^+^
Age		
<30 years old (ref)	1.00	1.00
≥30 years old	1.121 (0.715–1.756)	1.024 (0.587–1.786)
Marital Status		
Single (ref)	1.00	1.00
Married	**1.981** **(1.026–3.825)**	1.794 (0.826–3.896)
Other	0.615 (0.264–1.431)	0.612 (0.237–1.583)
Employment Status		
Not employed (ref)	1.00	1.00
Full-time employed	1.030 (0.598–1.773)	0.765 (0.397–1.473)
Other	0.857 (0.502–1.464)	0.654 (0.355–1.205)
Education		
Less than bachelor’s degree (ref)	1.00	1.00
Bachelor’s degree and higher	1.213 (0.779–1.888)	1.238 (0.721–2.125)
Household Income		
≤$50,000 (ref)	1.00	1.00
>$50,000	0.972 (0.618–1.527)	0.821 (0.495–1.361)
Insurance Status		
Uninsured (ref)	1.00	1.00
Insured	1.635 (0.892–3.000)	1.469 (0.743–2.902)
Chronic Disease Conditions		
None (ref)	1.00	1.00
At least one	1.496 (0.941–2.377)	**1.692** **(1.015–2.822)**

* Yes vs. no/not sure. ^+^ The adjusted models included all demographic variables listed in the table. Bold indicates *p* < 0.05.

## Data Availability

The data presented in this study are available on request from the corresponding author.

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
