# Peer review of "Usage of Digital Health Tools and Perception of mHealth Intervention for Physical Activity and Sleep in Black Women"

_ijerph, 2022, doi:10.3390/ijerph19031557_

Round 1

Reviewer 1 Report

The reviewed article provides a lot of data regarding the physical activity, sleep quality and personal perceptions of mHealth interventions (wearable trackers, feedback messages) of Black women.

The overall number of participants is sufficient. However, as the authors report themselves, the selection of the participants is biased, as they only used an online database. This way only persons with a good digital literacy take part in this study. This also means that some of the conclusions of the study “high perceived acceptability of remote interventions” have to be interpreted with care – as only persons who have registered themselves in a data base for persons interested in online research studies take part in this study.

I would propose that the authors either enlarge the number of participants to recruit further participants by offline ways or interpret the results regarding the perception of mHealth interventions with great care.

The numbers the authors present are quite interesting, but for the reader it would be more interesting to get insights into the why, what and how:

  1. The authors provide several references to demonstrate the greater prevalence of chronic health conditions or CVD risk factors among Black women. But no explanation was given why this is the case. A mHealth solution would be far more effective, if the underlying reasons for the chronic health conditions were known. I would propose to include literature references, which discuss these reasons.
  2. In table 1 the authors describe the demographic characteristics of the participants. It would be interesting to know how these characteristics differ from the average of the population in this age range. This could also help to further explain the underlying reasons for the greater prevalence of CVD risk factors among Black women.
  3. The most interesting part is the one about the usage of health-related apps, remotely delivered programs and feedback messages. However, it would be interesting to learn about the motivation and the acceptance of the participants (What type of data did they use (heart rate, steps, miles walked, sleep), what type of recommendation was useful for them (motivation, goals, training plans), would they accept further devices/ apps (nutrition apps, blood pressure meter, scale)) and even more about the impact of  these mHealth interventions (did they feel better after using the health app for some time, did the medical parameters improve?).

The result of this article is that a large part of the digital literate Black women owns a wearable or are willing to use one, already uses health apps and is interested to get personalized feedback on their activity level or to improve their sleep. This is an interesting outcome for health care professionals and providers of mHealth interventions, but it would be more valuable to learn about what the users of these health apps expect, accept (e.g. regarding data input) and need to have a positive impact in their life.

Author Response

The reviewed article provides a lot of data regarding the physical activity, sleep quality and personal perceptions of mHealth interventions (wearable trackers, feedback messages) of Black women.

Thank you for reviewing our paper and providing very thoughtful comments. We have made major changes to our introduction and discussion sections in response to your suggestions. We think your comments greatly helped us strengthening our paper.

Point 1: The overall number of participants is sufficient. However, as the authors report themselves, the selection of the participants is biased, as they only used an online database. This way only persons with a good digital literacy take part in this study. This also means that some of the conclusions of the study “high perceived acceptability of remote interventions” have to be interpreted with care – as only persons who have registered themselves in a data base for persons interested in online research studies take part in this study.

I would propose that the authors either enlarge the number of participants to recruit further participants by offline ways or interpret the results regarding the perception of mHealth interventions with great care.

Response 1: We appreciate this comment. We have added a paragraph in the discussion section that we hope helps the reader to interpret the study results. In this paragraph, we provide information about how similar/dissimilar our study sample is compared to available national data and other studies (see page 10, discussion section first paragraph). Also, in the limitations section, we have provided additional information about our study potentially being limited to Black women of reproductive age who have access to Internet and Prolific (please see page 12, first paragraph).

Point 2: The numbers the authors present are quite interesting, but for the reader it would be more interesting to get insights into the why, what and how:

The authors provide several references to demonstrate the greater prevalence of chronic health conditions or CVD risk factors among Black women. But no explanation was given why this is the case. A mHealth solution would be far more effective, if the underlying reasons for the chronic health conditions were known. I would propose to include literature references, which discuss these reasons.

Response 2: Thank you for your comment. We have revised the introduction section to include the reasons why Black women experience disparate rates of CVD. We explain the clinical factors associated with increased risk of CVD which impacts Black women. We also briefly introduce and explain structural racism as a fundamental driver of racial disparities in CVD. We cite the official statement on structural racism provided by the American Heart Association and provide citations that help to explain how structural racism inequitably shapes social and physical environments which in turn creates disparities in CVD, sleep, and physical activity. We want to make it clear to the reviewers, however, that the focus of our study was not to examine the impact of structural racism on CVD, physical activity, or sleep. But we highlighted in the paper that in our quest to develop mHealth interventions for Black women, these interventions must acknowledge and be responsive to the social context of Black women which is inextricably linked to structural racism. We emphasize in the paper that digital health/mHealth tools can help with reaching health equity for Black women (please see pages 1-2, introduction section paragraphs 2-3).

Point 3: In table 1 the authors describe the demographic characteristics of the participants. It would be interesting to know how these characteristics differ from the average of the population in this age range. This could also help to further explain the underlying reasons for the greater prevalence of CVD risk factors among Black women.

Response 3: Thank you for your comment on this. This is a great point. Aside from population health surveillance systems, other national surveillance systems that collect information about sociodemographics of the U.S. population (U.S. Census Bureau, BLS) often fail to report data in an intersectional way. As a result, it is difficult to find some data by race/ethnicity AND gender AND age group. We have tried our best to track down this often fragmented and incomplete national-level information to help the reader to get a sense of the representativeness of our sample. In the discussion section, we have added some data to help the reader get a sense of how our sample compares to the national scale (please see page 10, discussion section first paragraph).

Point 4: The most interesting part is the one about the usage of health-related apps, remotely delivered programs and feedback messages. However, it would be interesting to learn about the motivation and the acceptance of the participants (What type of data did they use (heart rate, steps, miles walked, sleep), what type of recommendation was useful for them (motivation, goals, training plans), would they accept further devices/ apps (nutrition apps, blood pressure meter, scale)) and even more about the impact of  these mHealth interventions (did they feel better after using the health app for some time, did the medical parameters improve?).

Response 4: We really appreciate this suggestion. We also believe these types of information will be very helpful for designing future mHealth interventions. Nevertheless, we think these types of information will be best obtained from qualitative approaches such as focus group studies and are beyond the scope of current study. We have now added this point to our discussion as a suggestion for future research direction (please see page 12, last paragraph).

Point 5: The result of this article is that a large part of the digital literate Black women owns a wearable or are willing to use one, already uses health apps and is interested to get personalized feedback on their activity level or to improve their sleep. This is an interesting outcome for health care professionals and providers of mHealth interventions, but it would be more valuable to learn about what the users of these health apps expect, accept (e.g. regarding data input) and need to have a positive impact in their life.

Response 5: We agree that this would be a very interesting future research direction. We have now added this point to our suggestion for future studies (please see page 12, last paragraph).

Reviewer 2 Report

This article presents data collected via an online questionnaire regarding their use of wearable sensors and perception of health-related apps that target physical activity and sleep.

The major issue with this research, as the authors themselves recognize, is sampling bias as the authors only recruited subjects via an online platform. The authors state that they were able to recruit subjects from a wide range of education and income level. That might be the case, but the question is if the subject distribution is representative of black women of childbearing age and if not, how the data should be interpreted. Prolific Academia enrolls subjects via social networking websites and word of mouth. Is there a subgroup of black women of childbearing age that might not have been represented or could have been possibly overrepresented in this study?  Please discuss.

It would be interesting to see the statistics for interaction between different factors, such as, age and education or age and income level. For example, authors state that “Women aged 30 years and older were less likely to have used any health-related app in the past 12 months compared to those who were younger” and I was wondering if this statement is also true for women older than 30 yrs who have higher level of education.

How does having insurance or the existence of chronic conditions influence the usage and perception of wearables? Same with marital or employment status?

Author Response

This article presents data collected via an online questionnaire regarding their use of wearable sensors and perception of health-related apps that target physical activity and sleep.

Thank you for taking your time to review our paper. We very much appreciate your helpful comments, in particular the ones about our statistical analyses. We believe our paper has improved significantly after incorporating your suggestions.

Point 1: The major issue with this research, as the authors themselves recognize, is sampling bias as the authors only recruited subjects via an online platform. The authors state that they were able to recruit subjects from a wide range of education and income level. That might be the case, but the question is if the subject distribution is representative of black women of childbearing age and if not, how the data should be interpreted. Prolific Academia enrolls subjects via social networking websites and word of mouth. Is there a subgroup of black women of childbearing age that might not have been represented or could have been possibly overrepresented in this study?  Please discuss.

Response 1: Thank you for your comment on this. Aside from population health surveillance systems, other national surveillance systems that collect information about sociodemographics of the U.S. population (U.S. Census Bureau, BLS) often fail to report data in an intersectional way. As a result, it is difficult to find some data by race/ethnicity AND gender AND age group. We have tried our best to track down this often fragmented and incomplete national-level information to help the reader to get a sense of the representativeness of our sample.  In the discussion section, we have provided some data to help the reader get a sense of how our sample compares to the national scale (please see page 10, discussion section first paragraph). Throughout the discussion, we have also described how our sample characteristics compare to other studies that used community-based samples which were often limited to one geographic area (e.g., Arizona, Floria).

Point 2: It would be interesting to see the statistics for interaction between different factors, such as, age and education or age and income level. For example, authors state that “Women aged 30 years and older were less likely to have used any health-related app in the past 12 months compared to those who were younger” and I was wondering if this statement is also true for women older than 30 yrs who have higher level of education.

Response 2: We really appreciate this comment.  In response to this comment and the comment below, we have re-run our data analysis to include results from both unadjusted and adjusted logistic regression. The adjusted logistic regression models included all demographic covariates: age, marital status, employment status, education level, household income, insurance status, and chronic disease conditions (see Table 2 – Table 4). To answer this specific comment, in the adjusted model, age was still a significant predictor of health-related app usage, controlling for education level. In other words, women aged 30 years and older were less likely to use any health-related app regardless of their education level. We have updated our results and discussion sections to reflect results from these unadjusted and adjusted models.

Point 3: How does having insurance or the existence of chronic conditions influence the usage and perception of wearables? Same with marital or employment status?

Response 3: Thank you for this suggestion. We have now examined the unadjusted and adjusted effects of insurance status, chronic disease conditions, marital status, and employment status on wearables ownership (Table 2), health-related app usages (Table 3), interest in remote interventions (Table 4) and personalized feedback messages (non-significant). We have updated our results and discussion sections to reflect results from these unadjusted and adjusted models (please see page 11 for discussion of these new results). In summary, employment status was associated with wearables ownership, employment and insurance status were associated with app usages, and chronic disease conditions was associated with interest in remote intervention.

Reviewer 3 Report

This manuscript reports the survey results performed in the convenience sample of Black women who were ready to respond to questions about physical activity, sleep patterns, and the use of wearable activity sensors.

The paper is rather straightforward, both in terms of the study design and performed analysis. The justification for the survey is not too convincing. The Introduction is superficial, and one can hardly understand the rationale for the study. It is not focused on the analyzed aspects. The explanation why physical activity and sleep are only examined and not other health behaviors is missing. Is nutrition less important in terms of the impact of cardiovascular health?

The study's objectives are stated indeed, but they are not justified as mentioned above in the Introduction. Promised analysis of associations is not provided in the manuscript.

There is no clear information about the structure of the applied questionnaire. It looks that no standard tools for assessing targeted outcomes have been used.

The Results section is as disappointing as other parts of the manuscript. It basically contains only simple frequencies without any ambition of actual analysis of the associations. A few colorful charts do not provide too much information and do not increase the value of the paper.

The Discussion is also shallow and weakly based on the literature. It is not clear what the basis of the drafted conclusions is.

References are very limited as for the topic addressed in the paper.

This paper decidedly does not conform to the quality level expected from the text published in the Q1 journal in public health.

Author Response

This manuscript reports the survey results performed in the convenience sample of Black women who were ready to respond to questions about physical activity, sleep patterns, and the use of wearable activity sensors.

Point 1: The paper is rather straightforward, both in terms of the study design and performed analysis. The justification for the survey is not too convincing. The Introduction is superficial, and one can hardly understand the rationale for the study. It is not focused on the analyzed aspects. The explanation why physical activity and sleep are only examined and not other health behaviors is missing. Is nutrition less important in terms of the impact of cardiovascular health?

Response 1: Thank you for your comments. We have revised our introduction section to strengthen its clarity and cohesion. We have also made revisions to help you to better understand the rationale for the study. Also, in the introduction, we have briefly explained the factors that are known to be associated with increased risk of cardiovascular disease, which of course includes nutrition. We have also provided language to make it clear that this study focuses on two of the many important factors associated with cardiovascular health—sleep and physical activity. We are not arguing nutrition is less important, however, dietary behavior is not within the scope of this study and is not the focus of the special issue that we were invited to be part of.

Point 2: The study's objectives are stated indeed, but they are not justified as mentioned above in the Introduction. Promised analysis of associations is not provided in the manuscript.

Response 2: Thank you for this comment. We indeed included results from our logistic regression models in the texts of result section. We recognize results presenting in the text-only form might be overlooked by readers. Therefore, we have now added 3 tables that show the results from our logistic regression models.

Point 3: There is no clear information about the structure of the applied questionnaire. It looks that no standard tools for assessing targeted outcomes have been used.

Response 3: As stated in our methods section, all survey items were derived from existing validated survey instruments or have been used in previous national studies, including the outcome measures (e.g., digital health tools usages). There are no standard measures for perceptions of mHealth interventions. Nevertheless, we used items from previous mHealth studies that examined similar constructs.

Point 4: The Results section is as disappointing as other parts of the manuscript. It basically contains only simple frequencies without any ambition of actual analysis of the associations. A few colorful charts do not provide too much information and do not increase the value of the paper.

Response 4: First, the original paper did include a series of logistic regressions that examined the association between digital health usage/perceptions and demographic variables. Further, based on suggestions from the other reviewers, we have now further strengthened our analyses by examining additional demographic variables and including results from adjusted models.

Point 5: The Discussion is also shallow and weakly based on the literature. It is not clear what the basis of the drafted conclusions is.

Response 5: Based on suggestions from the other reviewers, we have made major changes to the discussion section to (a) clearly present the key findings, (b) explain how our findings are consistent or inconsistent with prior work, and (c) explain the implications of our findings.

Point 6: References are very limited as for the topic addressed in the paper.

Response 6: Through our revision process, we have included additional relevant references (now a total of 59 references).

Point 7: This paper decidedly does not conform to the quality level expected from the text published in the Q1 journal in public health.

Response 7: We believe our paper has a unique contribution to the dearth of cardiovascular health-related mHealth literature focused on Black women of reproductive age in the United States. In particular, our topic was invited to be part of a special issue titled: The Ecologic Environment and Physical Activity: Innovations and Opportunities That Advance Research, Practice, and Policy. We think our paper is now significantly improved after incorporating suggestions from the other reviewers.

Round 2

Reviewer 3 Report

The manuscript has been considerably improved concerning the initial version provided for the first review.

The main improvements include:

  • extended Introduction explaining why the Authors emphasized the CVD risk of Black women
  • reformulated statement of the study aims
  • extended analysis of potential predictors associated with ownership of wearables, mobile health apps usage, interest in remotely delivered physical activity programs
  • extended Discussion
  • reformulated Conclusions
  • considerably increased references

Appreciating all amendments and extensions of the manuscript, there are still considerable limitations requiring efficient response and addressing in the paper before considering its publication.

They include:

  • Although the Introduction was significantly extended, we still do not know why the Authors focus only on physical activity and sleep and not other health behaviors in the context of cardiovascular health in Black women. Why have not other risks been included, or why only these two have been included? Are they any rationale for treating physical activity and sleep quality as the most important determinants of CV health in black women?
  • The survey should be thoroughly explained how the usage of specific mobile apps focused on physical activity and sleep quality has been addressed.
  • The Methods section does not provide information about the structure of the questionnaire; apart detailed description of its structure (type of items, number of items for specific aspects, response option), the questionnaire should be attached in the Supplementary file. Now, a reader does not really know what was assessed in the survey.
  • Furthermore, the Authors refer to ‘existing validated survey instruments’ without clear reference to these tools in specific areas covered in the questionnaire. The justification of using certain elements from earlier surveys should be indicated. Finally, if the Authors have not applied validated tools, it should be transparently stated.

From minor issues, it is not clear why the Authors present data adding to 100% as a bar graph and not as a pie chart. Anyway, such data could be provided in a table including percentages and absolute frequencies.

Author Response

The manuscript has been considerably improved concerning the initial version provided for the first review.

The main improvements include:

  • extended Introduction explaining why the Authors emphasized the CVD risk of Black women
  • reformulated statement of the study aims
  • extended analysis of potential predictors associated with ownership of wearables, mobile health apps usage, interest in remotely delivered physical activity programs
  • extended Discussion
  • reformulated Conclusions
  • considerably increased references

Thank you for your positive comments for our revision.

Appreciating all amendments and extensions of the manuscript, there are still considerable limitations requiring efficient response and addressing in the paper before considering its publication.

They include:

Point 1: Although the Introduction was significantly extended, we still do not know why the Authors focus only on physical activity and sleep and not other health behaviors in the context of cardiovascular health in Black women. Why have not other risks been included, or why only these two have been included? Are they any rationale for treating physical activity and sleep quality as the most important determinants of CV health in black women?

Response 1: We appreciate this comment.  We have now added a few more sentences in our introduction to more clearly explain our focus on physical activity and sleep in this paper:

“While we recognize all these factors have their unique contributions to a person’s CVD risk, this paper will focus on two of the important modifiable behavioral factors: physical activity and sleep in the scope of implications for digital and mobile health (mHealth) interventions. Physical activity and sleep represent two daily behaviors that can be automatically and continuously tracked by a wearable sensor, thus providing opportunities to develop personalized, adaptive behavioral interventions.”

Point 2: The survey should be thoroughly explained how the usage of specific mobile apps focused on physical activity and sleep quality has been addressed.

Response 2: We have now included our survey as a supplemental material. Thank you.

Point 3: The Methods section does not provide information about the structure of the questionnaire; apart detailed description of its structure (type of items, number of items for specific aspects, response option), the questionnaire should be attached in the Supplementary file. Now, a reader does not really know what was assessed in the survey.

Response 3: Thank you. We have now included our survey as a supplemental material.

Point 4: Furthermore, the Authors refer to ‘existing validated survey instruments’ without clear reference to these tools in specific areas covered in the questionnaire. The justification of using certain elements from earlier surveys should be indicated. Finally, if the Authors have not applied validated tools, it should be transparently stated.

Response 4: Thank you for this comment. We have now added a sentence to explain why we didn’t use the full scale.

“Since the primary purpose of the survey is not to obtain a complete picture of participants’ behaviors (e.g., usual physical activity and sleep patterns), we used items from validated surveys to capture a “snapshot” of their current physical activity and sleep quality while keeping the survey brief.”

Point 5: From minor issues, it is not clear why the Authors present data adding to 100% as a bar graph and not as a pie chart. Anyway, such data could be provided in a table including percentages and absolute frequencies.

Response 5: Thank you for this comment. The data does not add to 100% because the response options were “check all that applied.” We have now included the survey as supplemental material, which will help to clarify this point.